# Structural basis of drug recognition by human MATE1 transporter

Ksenija Romane ⬤[1], Giulia Peteani[2,3], Somnath Mukherjee ⬤[4], Julia Kowal ⬤[1], Lorenzo Rossi ⬤[1], Jingkai Hou ⬤[4], Anthony A. Kossiakoff ⬤[4], Thomas Lemmin ⬤[2] & Kaspar P. Locher ⬤[1] ✉

Human MATE1 (multidrug and toxin extrusion protein 1) is highly expressed in the kidney and liver, where it mediates the final step in the excretion of a broad range of cationic drugs, including the antidiabetic drug metformin, into the urine and bile. This transport process is essential for drug clearance and also affects therapeutic efficacy. To understand the molecular basis of drug recognition by hMATE1, we determined cryo-electron microscopy structures of the transporter in complex with the substrates 1-methyl-4-phenylpyridinium (MPP) and metformin and with the inhibitor cimetidine. The structures reveal a shared binding site located in a negatively charged pocket in the C-lobe of the protein. We functionally validated key interactions using radioactivity-based cellular uptake assays using hMATE1 mutants. Molecular dynamics simulations provide insights into the different binding modes and dynamic behaviour of the ligands within the pocket. Collectively, these findings define the structural basis of hMATE1 substrate specificity and shed light on its role in drug transport and drug-drug interactions.

Excretion of drugs and waste products is a fundamental physiological process carried out primarily by the liver and kidneys. This process depends on the concerted action of membrane transporters localized to the apical and basolateral sides of hepatocytes and renal tubular cells[1]. A key player in this vectorial transport system is the multidrug and toxin extrusion transporter family (MATE, SLC47A). In humans, the MATE family comprises two isoforms, hMATE1 and hMATE2-K[2]. hMATE1 is mainly expressed in the apical membrane of renal tubular epithelial cells and in the canalicular membrane of hepatocytes, where it mediates the final excretion step of drugs and endogenous metabolites into urine and bile. hMATE2-K, in contrast, is expressed specifically in the kidney. In the renal proximal tubule, hMATE1 works in tandem with the basolateral organic cation transporter OCT2 to mediate the secretion of cationic substrates into the urine[3]. In the liver, hMATE1 facilitates biliary excretion following uptake by OCT1. Due to this coordinated role in drug elimination, hMATE1 shares considerable substrate specificity with OCTs. Physiological substrates of hMATE1

include organic cations such as creatinine, guanidine, thiamine, and the organic anion estrone sulfate. In addition, hMATE1 transports several clinically important drugs, including metformin, cimetidine, oxaliplatin and acyclovir[4].

Numerous in vivo studies have demonstrated the crucial role of MATE transporters in the systemic elimination of drugs. Studies in mice have shown decreased renal clearance of metformin[5] and cephalexin[6] in knockout Mate1 models. Furthermore, genetic deletion or pharmacological inhibition of MATE1 during cisplatin treatment in mice led to markedly elevated renal accumulation of the chemotherapeutic agent and aggravated nephrotoxicity[7]. Among MATE1 substrates, metformin is of particular clinical interest, as it is widely prescribed as a first-line treatment for type 2 diabetes mellitus. Studies in humans have shown that administration of the antihistamine cimetidine, an $H_2$ receptor antagonist, significantly reduces the overall renal clearance of metformin[8]. Subsequent in vitro studies have demonstrated that this drug-drug interaction is

[1]Institute of Molecular Biology and Biophysics, ETH Zürich, Zürich, Switzerland. [2]Institute of Biochemistry and Molecular Medicine, Universität Bern, Bern, Switzerland. [3]Graduate School for Cellular and Biomedical Sciences (GCB), University of Bern, Bern, Switzerland. [4]Department of Biochemistry and Molecular Biology, The University of Chicago, Chicago, IL, USA. ✉e-mail: locher@mol.biol.ethz.ch

mediated by hMATE1 in renal epithelial cells, where cimetidine acts as a potent competitive inhibitor of the transporter[9,10]. Given the effect of human MATE1 on the pharmacokinetics of therapeutic drugs, the U.S. Food and Drug Administration (FDA)[11] and the European Medicines Agency (EMA)[12] recommend evaluating new drug candidates for their inhibitory effects on MATE1 and MATE2-K in in vitro assays.

Despite their clinical relevance, the mechanism of substrate transport by human MATEs is not completely understood. MATE transporters are conserved across all domains of life and mediate the efflux of structurally and chemically diverse metabolic and xenobiotic compounds by coupling substrate export to an oppositely directed ion gradient[13]. In humans, MATE transporters utilize the proton gradient as the driving force for substrate extrusion[9,14]. Some bacterial homologues, such as NorM from *Vibrio parahaemolyticus*, are sodium-coupled[15], while others, such as NorM from *Vibrio cholerae*, are driven by both sodium and proton gradients[16]. Structural studies of bacterial MATEs have revealed a conserved fold characteristic of the MATE family, consisting of 12 transmembrane helices organized in two lobes (N- and C-terminal lobes) that are related by pseudosymmetry[17,18]. The topology of these helices differs from that of other 12-transmembrane transporter families, such as the major facilitator superfamily (MFS)[13,19]. In addition to this conserved core, human MATE1 contains an additional transmembrane helix (TM13), the function of which is still unknown. Truncation of this helix in rabbit MATE1 does not abolish transport activity, but significantly reduces transport levels, while substrate specificity remains unaffected[20]. To date, no structures of hMATE1 have been reported. The only available eukaryotic MATE structures – CasMATE from *Camelina sativa*[21] and AtDTX14 from *Arabidopsis thaliana*[22] – were solved in the apo state, leaving the location of the substrate-binding site in eukaryotic MATEs uncharacterized. Homology models of hMATE1 have been generated based on available structures from VcNorM[23,24] and MATE from *Pyrococcus furiosus* (PfMATE)[24], predicting the substrate-binding site in the central cavity or N-lobe. In contrast, homology models based on the eukaryotic CasMATE[25] template have placed the substrate-binding site in the C-lobe of the transporter.

In this work, we present cryo-EM structures of human MATE1 both in the apo state and in complex with its substrates metformin and 1-methyl-4-phenylpyridinium (MPP[+]), as well as the inhibitor cimetidine. The structures reveal key interactions in the binding pocket involved in substrate recognition, which we validate through functional assays. Molecular dynamics simulations further elucidate the dynamic behavior of these ligands within the pocket.

## Results

### Structure determination of hMATE1

Prior to structural studies, we confirmed the transport activity of human MATE1 expressed in HEK293 cells. By establishing a proton gradient to drive transport[26], we measured the cellular uptake of [3H]-MPP[+], a prototypical organic cation and well-characterized hMATE1 substrate. We further demonstrated the selectivity of hMATE1 for different substrates by examining the uptake of the fluorescent dye 4′,6-diamidino-2-phenylindole (DAPI) (Fig. 1c, d)[27].

The relatively small size of hMATE1 (62 kDa) hindered high-resolution structure determination by cryo-EM. To increase particle size and conformational stability, we generated synthetic antigen-binding antibody fragments (Fabs) conformationally specific for hMATE1. Among them, MATE1_Fab 3 and MATE1_Fab 6, were selected for further characterization. We tested the effect of these Fabs in a DAPI uptake assay using hMATE1-expressing cells (Fig. 1c). Incubation with MATE1_Fab 6 significantly reduced intracellular DAPI fluorescence, indicating the inhibitory effect of this Fab, whereas MATE1_Fab 3 did not inhibit transport activity. We used both Fabs for cryo-EM structure determination of hMATE1. We purified and reconstituted hMATE1 into nanodiscs, creating a native-like lipid environment. We

then added the corresponding Fab, which increased the size of the complex (Supplementary Fig. 1). Using this strategy, we determined cryo-EM structures of hMATE1 in the absence of ligands (apo state) at 2.95 Å resolution with MATE1_Fab 6 and at 3.7 Å with MATE1_Fab 3 (Fig. 2a, Supplementary Figs. 2, 3). Both Fabs bind to the extracellular side of the transporter. For MATE1_Fab 6 most of the interactions are mediated by the longest loop, CDR-H3 (complementarity-determining region 3 of the heavy chain), which interacts with both the N- and C-lobe, while CDR-H1 and CDR-H2 further stabilize the complex (Fig. 2c). MATE1_Fab 3 interacts with the N-lobe with the three CDRs of the heavy chain and has only a single amino acid contact (Y101 of CDR-H3 with F282 of hMATE1) with the C-lobe. Both Fab-bound structures revealed the same outward-open conformation of the transporter with a root mean square deviation (RMSD) of 0.7 Å for hMATE1 (Fig. 2b). This confirmed that MATE1_Fab 6 binding does not distort the protein structure but instead locks a native conformational state, supporting the physiological relevance of the Fab-bound structures. Since both Fabs trap the same conformation, we infer that both Fabs are compatible with substrate binding, while MATE1_Fab 6 blocks the substrate release.

### Structures of substrate- and inhibitor-bound hMATE1

We selected MATE1_Fab 6 for further studies as it allowed for higher resolution structure determination. Using this Fab, we determined high-resolution cryo-EM structures of hMATE1 in complex with two different substrates and an inhibitor (Fig. 3, Supplementary Figs. 4–8): hMATE1-metformin at 2.3 Å, hMATE1-MPP at 3.1 Å and hMATE1-cimetidine at 3.3 Å. The core structure of hMATE1 corresponds to the classical MATE fold[13,19], displaying pseudosymmetry between the two halves of the transporter – the N-lobe (TMs 1-6) and the C-lobe (TMs 7-12) (Fig. 3b, c). hMATE1 contains an additional transmembrane helix, TM13, which is connected to TM12 by a long intracellular loop. This loop was not resolved in our structure, likely due to its flexibility. The transporter adopts an outward-open conformation, with the central cavity between the N- and C-lobes exposed to the extracellular space. Within this cavity and in close proximity (~4 Å) to the CDR-H3 loop of the Fab, we observed an extra density. Based on the characteristic shape of the density, we modeled it as a cholesterol molecule (Supplementary Fig. 9). The presence of cholesterol likely arises from the nanodisc reconstitution, where we used a defined mixture of brain polar lipid extract supplemented with cholesterol. To our knowledge, the role of cholesterol in MATE1 has not been previously investigated, and any potential functional relevance remains to be determined.

Due to the low affinity ($K_m = 0.1$ mM)[28] of the transporter for MPP, we added 1.5 mM of this compound to our hMATE1-Fab6 complex. The hMATE1-MPP cryo-EM map featured a well-defined density in the C-lobe pocket of hMATE1 (Fig. 3a), which was absent in the apo state. We also determined cryo-EM structures in complex with the anti-diabetic drug metformin and the inhibitor cimetidine. For structure determination, we used 1.2 mM cimetidine and 10 mM metformin, concentrations that are ~10-fold above the reported $K_m$ of these compounds[28]. The cryo-EM maps of hMATE1-metformin and hMATE1-cimetidine revealed extra densities of distinct shape in the same C-lobe pocket as for MPP (Fig. 3a, b). The overall structures of hMATE1-apo and ligand-bound states showed the same outward-facing conformation, and the RMSD between the apo and ligand-bound states was 0.3–0.4 Å. (Fig. 3b).

The substrate-bound state did not show any major conformational changes with respect to the apo state. The only difference in the binding pocket was in the rotamer of W274 (Fig. 3a). In the apo structure, W274 was only partially resolved, while the rest of the substrate-binding pocket was well defined. At a lower threshold, density corresponding to two different rotamers was observed (Fig. 3a), which could be explained by the conformational flexibility of this residue. Therefore, we modeled two possible rotamers for W274 (W274-1 and W274-2)

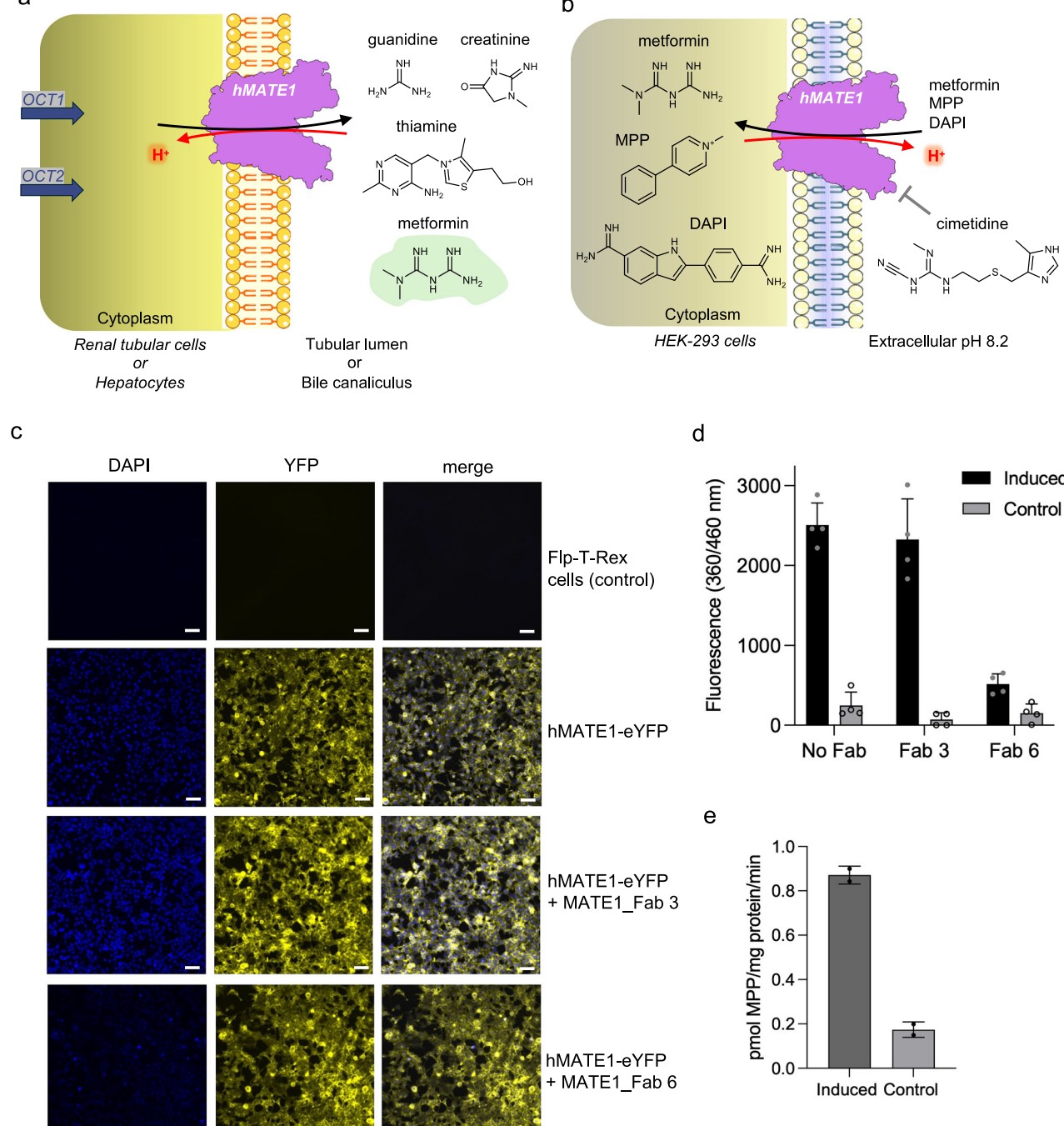

**Fig. 1 | Human MATE1 function. a** hMATE1 role in the renal tubular cells and hepatocytes of exporting physiological substrates or clinically used drugs (e.g. metformin, highlighted in green). **b** in vitro assay setup in HEK293 cells. $^3$H-MPP$^+$ was used for uptake mediated by hMATE1 and detected by radioactivity (See **e**). DAPI uptake was detected by fluorescence (See **c**, **d**). Metformin and cimetidine (inhibitor) were used for competition assays of MPP uptake (See Fig. 6). **c** Fluorescent microscopic images showing stained nuclei in blue (uptake of DAPI) and expression of hMATE1 in yellow (co-expression of hMATE1 with eYFP protein) after incubation of the inducible stable cell line of hMATE1 with DAPI for 10 min at pH 8.2.

Pre-incubation with MATE1_Fab 3 does not affect the uptake, while MATE1_Fab 6 inhibits it. Scale bar represents 100 μm. **d** DAPI uptake was quantified by measuring fluorescence intensity at excitation of 360 nm and emission of 460 nm. Values are shown as mean ± SD from $n = 4$ biological replicates. **e** Uptake of $^3$H-MPP$^+$ in tetracycline-induced stable cell line of hMATE1 for 10 min and pH 8.4 compared to cells not overexpressing hMATE1 (Flp-T-Rex cells). Values are shown as mean ± SD from $n = 2$ biological replicates. Source data are provided as a Source Data file. Images in panels (**a**) and (**b**) provided by Servier Medical Art (https://smart.servier.com/), licensed under CC BY 4.0 (https://creativecommons.org/licenses/by/4.0/).

in the apo state. In contrast, a strong density of W274 was observed in the substrate-bound state, corresponding to rotamer W274-2 (Fig. 3a).

### Substrate binding pocket of hMATE1

We found that the substrate binding pocket of hMATE1 is located halfway through the membrane, within the C-lobe of the transporter.

This pocket is negatively charged (Fig. 4), consistent with its role in accommodating cationic substrates. It is lined by two glutamates (E273 and E389) and aromatic residues (Y277, Y299, Y306 and W274) (Fig. 4).

Next, we analyzed how individual ligands interact with key residues within this site. Metformin has a pKa of 11.5 and has a positive charge at physiological pH. The partial positive charge of the amino

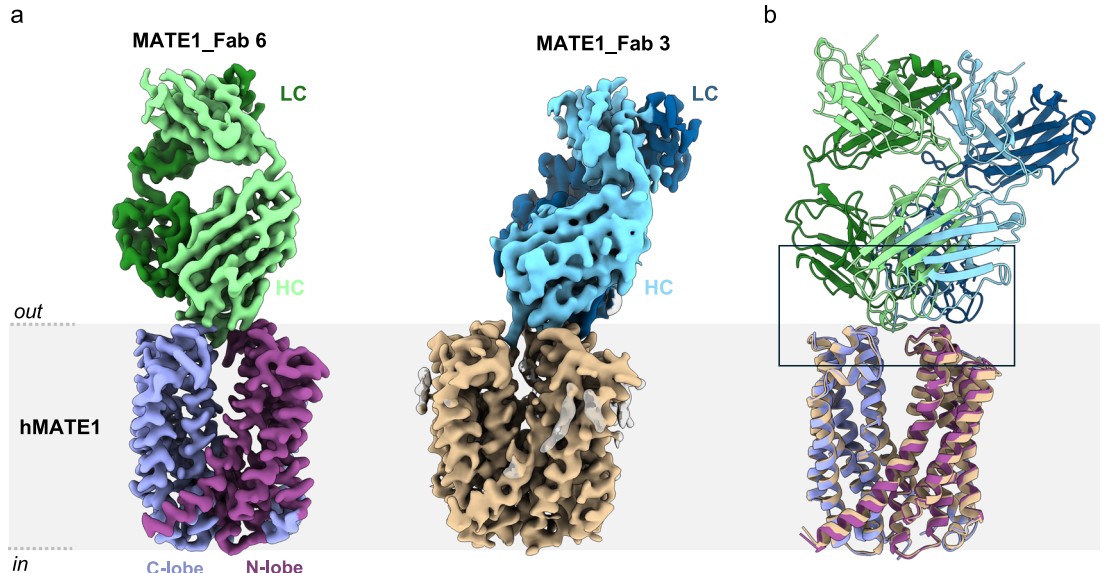

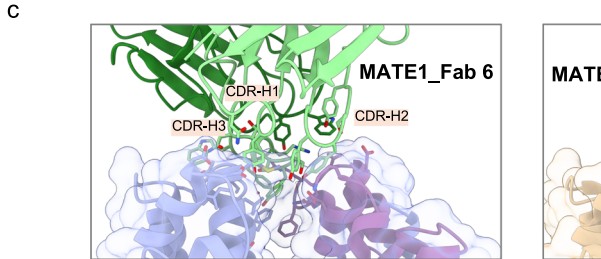

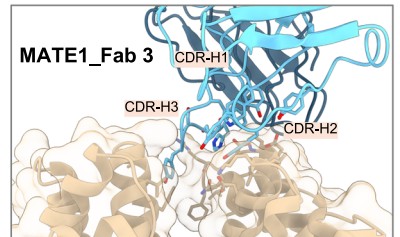

**Fig. 2 | Apo structures of hMATE1 with two different Fabs. a** Cryo-EM maps of the hMATE1 in complex with MATE1_Fab 6 (2.95 Å) and MATE1_Fab 3 (3.7 Å). **b** Superposition of the two models shows the different binding epitopes of the Fabs. **c** Detailed view of the binding epitope for each Fab.

groups allows them to form Coulomb interactions with E273 and E389. In addition, metformin forms cation-π interactions with neighboring aromatic residues Y277 and Y306 (Fig. 4a, b). Initial evaluation of the cryo-EM map suggested that, at a lower threshold, the density could accommodate two molecules of metformin (Fig. 3a, Supplementary Fig. 10). We used molecular dynamics (MD) simulations to assess whether two metformin molecules can simultaneously occupy the binding pocket. In simulations where two molecules were initially placed in the pocket, one rapidly diffused out within the first 150 ns (Supplementary Movie 1), suggesting that stable binding of two molecules is unlikely. To further explore the conformational landscape, we performed MD simulations starting with a single metformin molecule, modeled within the cryo-EM density observed in the binding pocket. Additionally, metformin was docked with Swissdock[29] and the top five poses were also used for MD simulations. In all cases, metformin showed high mobility within the binding pocket and all variants sampled a similar conformational space. These simulations also revealed that the binding pocket provides sufficient space for metformin to reorient itself (Supplementary Movie 2). A single molecule model exhibiting conformational flexibility within the binding pocket is consistent with our cryo-EM map, which reveals an extra density at lower thresholds (Fig. 3a). This observation is further in agreement with its low affinity as a substrate.

The cryo-EM density of cimetidine, combined with MD simulations of two possible binding orientations (Supplementary Fig. 10), allowed us to place the inhibitor in the substrate-binding pocket of hMATE1. One of the poses was unstable and dissociated during the simulation (Supplementary Movie 3), while the other remained stably

bound (Fig. 4c, Supplementary Movie 4). In this configuration, cimetidine forms a hydrogen bond via the NH group of its methyl-imidazole moiety with E389. In addition, the methyl-imidazole ring forms π-π stacking interactions with Y277 and Y306 (Fig. 4d). The cyanoguanidine group contributes to the binding through hydrophobic interactions with W274, N82 and I303. In contrast to the significant mobility of metformin within the binding pocket, the methyl-imidazole ring of cimetidine appeared to be "locked" between Y277 and Y306, forming π-π interactions. In comparison, the cyanoguanidine moiety displayed great flexibility within the binding pocket (Fig. 5b), reflecting different interaction patterns of the two ends of the molecule.

We could confidently model an MPP molecule inside the density. The cationic 1-methylpyridinium group points toward E389, with the charged nitrogen located ~6 Å away from this residue. Both aromatic rings of MPP engage in π-π stacking interactions with W274, along with hydrophobic contacts with Y299, I303 and Y306 (Fig. 4f). MD simulations of MPP (4 × 600 ns) from our experimentally determined position provided further insight into the interactions and dynamic behavior of this molecule inside the binding pocket (Supplementary Movie 5). These simulations also revealed π-π stacking interactions between the methylpyridinium ring of MPP and Y277 (Fig. 5d).

**Functional analysis of binding pocket residues**

Based on our structures, we identified six amino acids in close proximity to metformin, cimetidine and MPP (E273, W274, Y277, Y299, Y306, E389) and generated single point mutants of hMATE1, which we expressed in HEK293 cells. We verified the membrane expression of each mutant by fluorescence microscopy (Supplementary Fig. 11). All

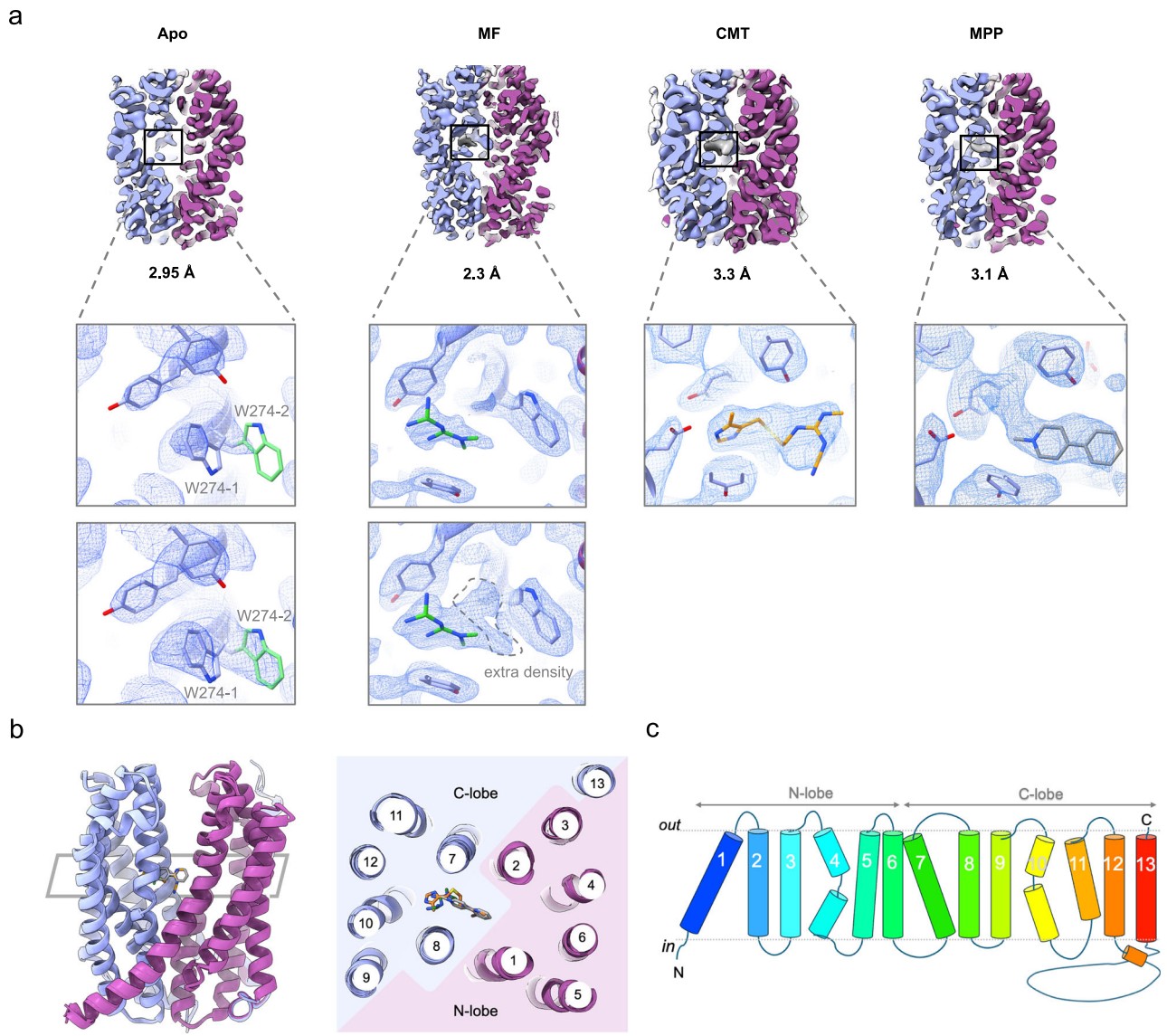

**Fig. 3 | Structures of hMATE1 in apo and substrate-bound states. a** Cryo-EM maps of hMATE1 in apo state and in complex with metformin (MF), cimetidine (CMT) and MPP. All of the complexes were determined with MATE1_Fab 6. Fab density is omitted for clarity. Vertical slice through the transporter (top panel) shows ligand density (grey) in the binding pocket. Close-up view (bottom panel) shows the cryo-EM map (blue mesh) around the bound ligands. Note that the close-up views are not identically oriented to better illustrate the ligand density. In the apo state, density corresponding to two rotamers of W274 is observed. In the MF-bound state, extra density around metformin density is observed at a lower threshold. **b** Overlayed structures of hMATE1 (apo and ligand-bound) on the left and view from the extracellular side on the right. **c** Topology of hMATE1.

of them localized to the plasma membrane, although their expression levels varied. To gain insight into the interaction of individual residues of the binding pocket with MPP, we performed MPP uptake assay for all the mutants in HEK293 cells (Fig. 6a). To account for variable plasma membrane expression, uptake values were normalized to the relative surface expression of each mutant (Supplementary Fig. 12).

MPP uptake was slightly affected by the Y306A and Y299A substitutions, resulting in a ~20% decrease and ~10% increase, respectively. In contrast, the Y299F mutation resulted in a 1.7-fold increase in uptake, possibly by providing additional hydrophobic interactions with the phenyl group of MPP. This suggests that Y299 and Y306 are not involved in MPP binding or that their roles can be compensated by other aromatic residues in the binding pocket. In contrast, mutation of Y277A completely abolished the transport activity, which was partially restored (~50%) by the Y277F mutation, highlighting the requirement for an aromatic side chain at this position. Supporting this, MD simulations suggest that Y277 forms π-π stacking interactions with MPP.

Mutations of glutamates E273 and E389 and tryptophan W274 abolished MPP uptake activity (Fig. 6a). Previous reports have shown that mutation of E273, E389 and W274 impaired transport of MPP or tetraethylammonium (TEA)[14,22,30]. The two glutamates are well conserved across MATEs in different organisms and have been proposed as the proton binding site for eukaryotic MATEs[22,30]. In addition to the glutamates, W274 is also highly conserved in eukaryotic MATEs[22]. Studies of plant MATEs have proposed that W274 plays a role in occluding hydrogen bonding of the conserved glutamates in the pocket[22]. Our cryo-EM structures suggest that the main W274 rotamer (W274-1) observed in the apo state may serve a similar function by shielding the binding pocket from the solvent-accessible central cavity.

The influence of selected mutations on cimetidine and metformin transport was indirectly evaluated using competition assays. We measured [3]H-MPP[+] uptake in the presence of increasing concentrations of unlabeled metformin and cimetidine (Fig. 6c, d). Metformin

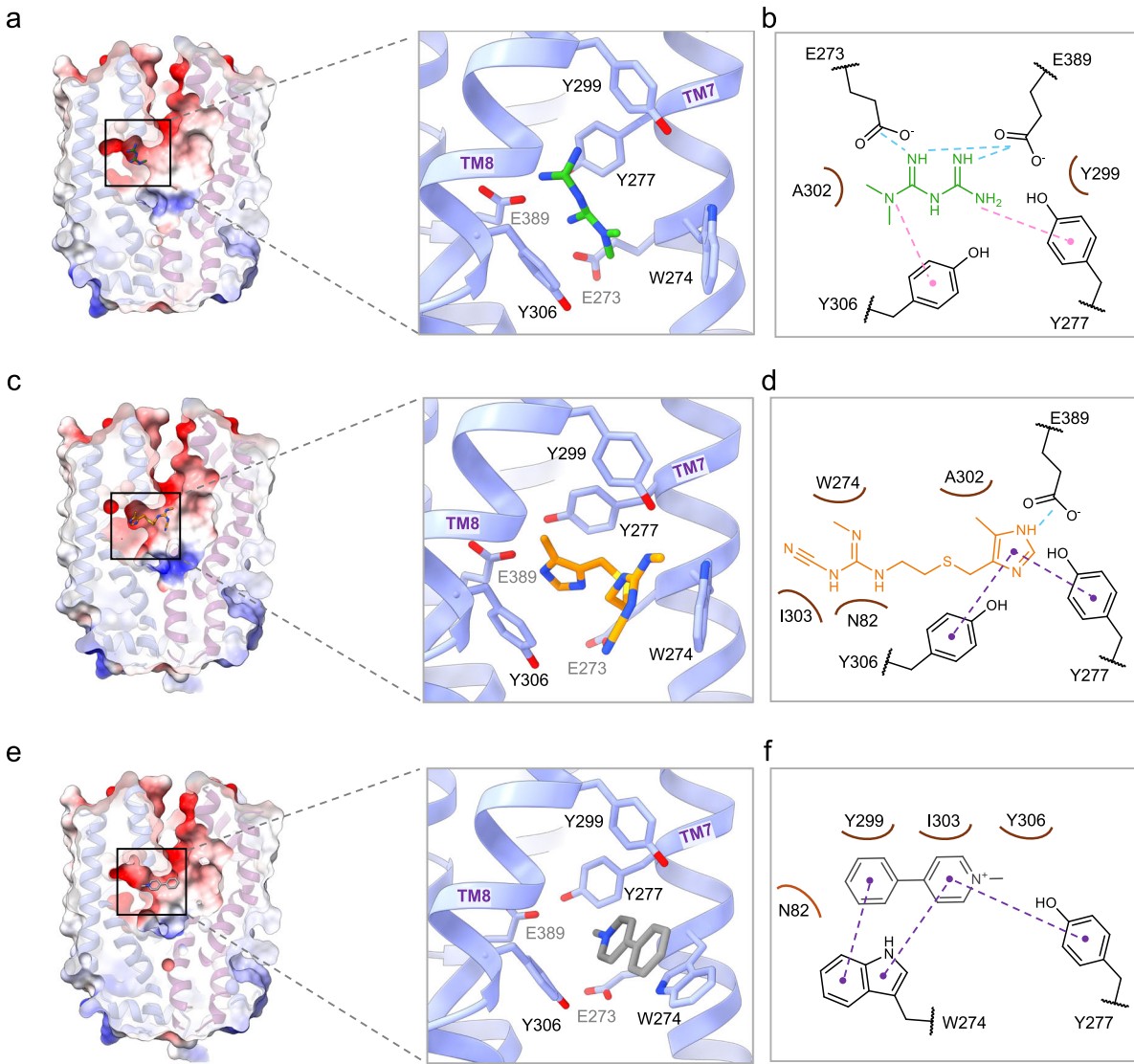

**Fig. 4 | Substrate binding pocket of hMATE1. a** Vertical slice through an electrostatic surface representation of hMATE1, red to blue: −10 kcal/(mol·e) to +10 kcal/(mol·e), showing metformin in the substrate binding pocket. Right panel shows a close-up view of the substrate binding pocket indicating the residues within 4 Å distance of metformin. **b** Detailed interactions between metformin and hMATE1 residues. Cation-π interactions are shown as pink and polar interactions are shown as blue dashed lines. Hydrophobic interactions are shown as arcs. **c**, **d** same as **a**, **b** for cimetidine. π-π interactions are shown as purple dashed lines. **e**, **f** same as **a**, **b** for MPP. π-π interactions between MPP and Y277 were added based on the MD simulations data (Fig. 5d).

and cimetidine inhibited MPP uptake via hMATE1 in a dose-dependent manner. In our assays, metformin exhibited an $IC_{50}$ inhibition constant of 58 ±11 μM, consistent with previous reports in CHO cells of 47 μM[26]. In contrast, cimetidine showed significantly stronger inhibition, with an $IC_{50}$ of 2.7 ±0.7 μM, in agreement with the 2.7 μM previously measured in other studies[10]. Kinetic studies of metformin uptake have shown that cimetidine acts as a competitive inhibitor of metformin[10].

For metformin, all mutants exhibited a clear shift in the $IC_{50}$ values compared to wild-type (Fig. 6d). The most prominent differences were observed for Y299A, W274F and Y306A. The effect of Y306 on metformin transport is explained by its involvement in cation-π interactions with metformin. In contrast, the Y306F mutation had an opposite effect in the context of cimetidine, resulting in a decreased $IC_{50}$. This suggests that the π-π interactions with cimetidine are more favorable without the hydroxyl group of the tyrosine at this position. However, Y306A had no effect on cimetidine competition, suggesting that this residue is not critical and can be compensated by other interactions in the pocket. As for MPP, Y306 did not have a major effect on the uptake

(Fig. 6a). This indicates a differential involvement of this residue depending on the substrate.

The W274 mutant exhibited a clear shift in the $IC_{50}$ of cimetidine and metformin (4–10× fold), supporting the important role of tryptophan in the binding pocket for all substrates (Fig. 6d). For cimetidine, the W274A mutation had the most notorious effect on activity. We observed in MD simulations that the cyanoguanidine moiety interacts with W274, which could be critical for cimetidine transport.

Because E273 and E389 mutants virtually abolished MPP uptake activity, they were excluded from our competition assays (Fig. 6a). However, previous studies have shown that mutation of these residues impairs the transport of cimetidine and other substrates of hMATE1[22,30]. Furthermore, these residues interact with both cimetidine and metformin (Fig. 4a, c), providing the necessary negatively charged environment for the binding of the cationic substrates of hMATE1. We therefore propose that these two glutamates are also critical for the recognition and transport of other hMATE1 substrates.

The variation in how individual mutations affect metformin and cimetidine inhibition profiles suggests different modes of substrate

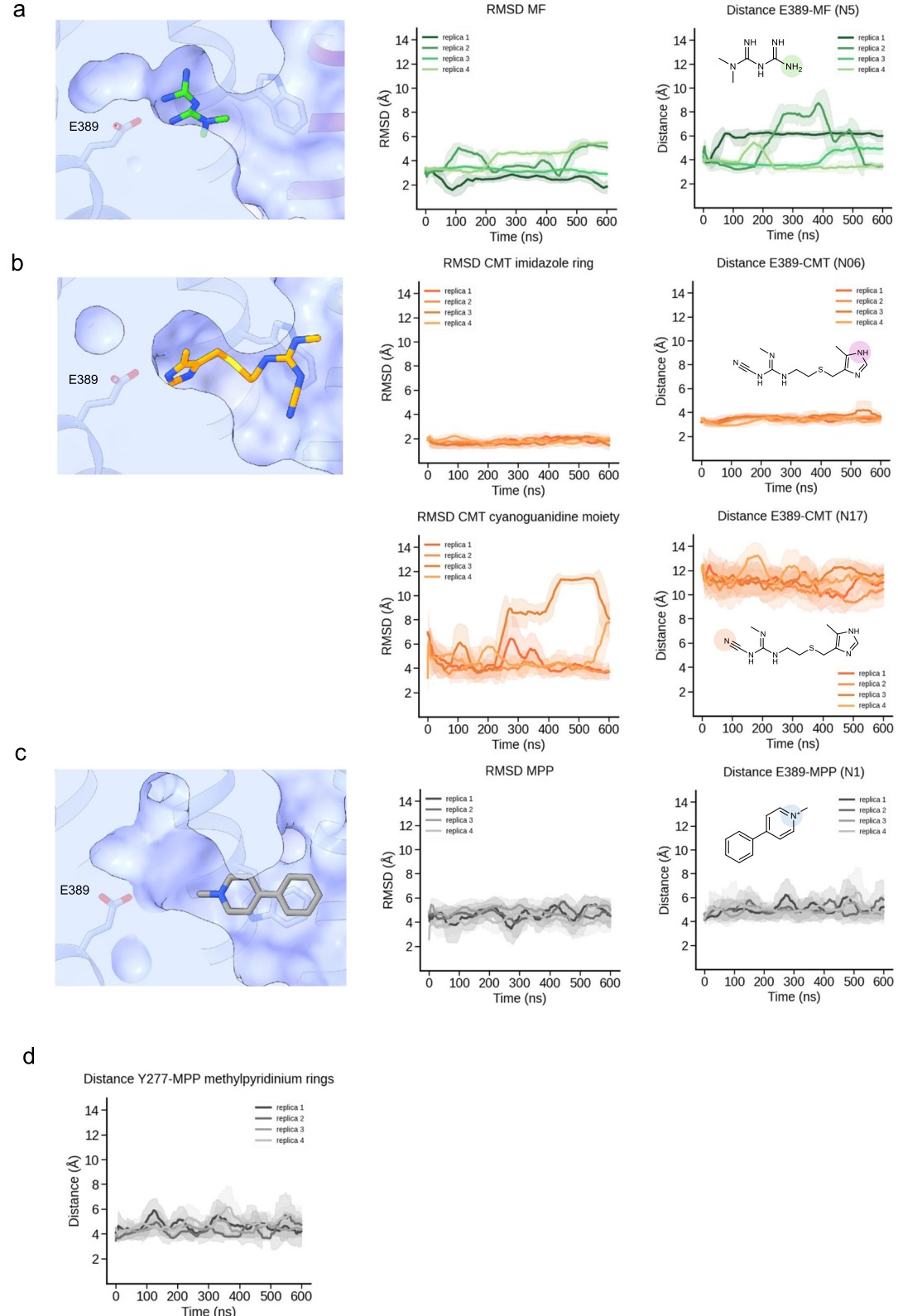

**Fig. 5 | Molecular dynamics simulations of substrates and inhibitor inside the binding pocket. a–c** Initial position of the ligand (metformin, MPP and cimetidine respectively) inside the binding pocket at the beginning of simulation (left). RMSD of each ligand during molecular dynamics simulations performed for four replicas (center). The plot shows fluctuations in ligand position relative to the initial state, indicating its degree of mobility inside the binding pocket. For cimetidine, two plots are shown to reflect the different degree of mobility of its methyl-imidazole ring compared to the cyanoguanidine moiety. Distance between E389 and selected atoms of ligands (highlighted in the chemical structures) during molecular dynamics simulations (right). **d** Distance between Y277 and methylpyridinium ring of MPP during molecular dynamics simulations. For all the plots distances are shown as rolling averages (window size = 50 frames) with shaded areas representing the rolling standard deviation.

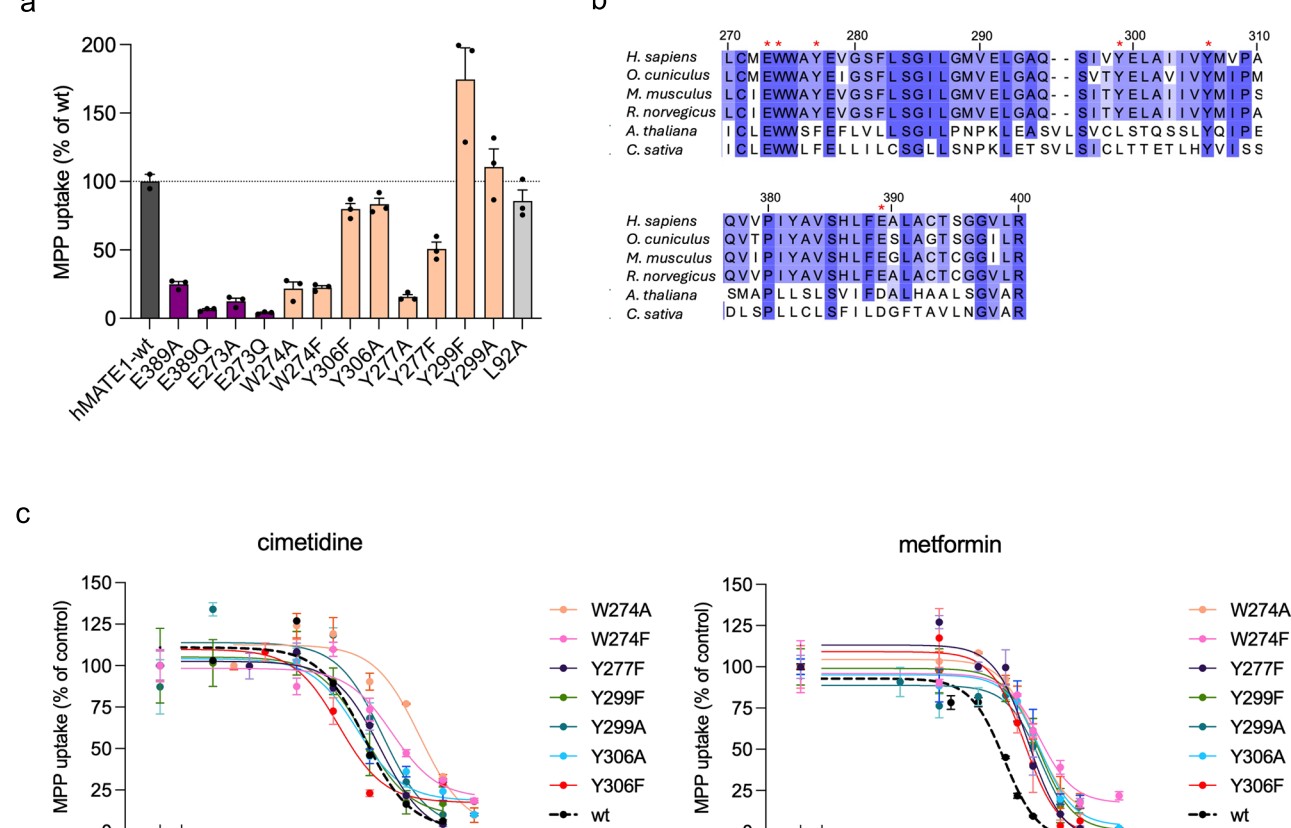

**Fig. 6 | Effects of hMATE1 mutants on its transport activity. a** Uptake of ³H-MPP⁺ into HEK293 cells expressing hMATE1 mutants. Activities are expressed relative to that of the wild-type (wt, 100% activity) and normalized to the membrane protein expression (Supplementary Fig. 12). Mutants of glutamate residues in the substrate binding pocket are shown in purple and of aromatic residues in beige. Control mutant is shown in grey. Bars represent means of *n* = 3 biological replicates, error bars represent the S.E.M. **b** Multiple sequence alignment of MATE1 proteins from mammalian and plant species generated using Clustal Omega. Mutated residues are marked with asterisks. **c** Cold-competition inhibition of ³H-MPP⁺ transport by cimetidine (left) and metformin (right). Activities are expressed relative to the control, i.e. in the absence of cimetidine or metformin. Each point represents the mean value of *n* = 3 biological replicates and error bars represent the S.E.M. **d** Half-inhibition concentration (IC₅₀) estimated by nonlinear regression analysis of the data shown in (**c**). Source data are provided as a Source Data file.

recognition by hMATE1. MD simulations further explain the higher inhibitory potency of cimetidine compared to metformin, as observed in our competition assays. The positively charged MPP enters the binding pocket and interacts mainly with Y277, while exhibiting high mobility inside the binding pocket due to its small size. The bulkier cimetidine accommodates inside the same binding pocket with its methyl-imidazole moiety "locked" between Y277 and Y306 and stabilized by a hydrogen bond to E389, while its cyanoguanidine group forms additional hydrophobic interactions with surrounding residues. This way it blocks the entry of MPP and its possible recognition by

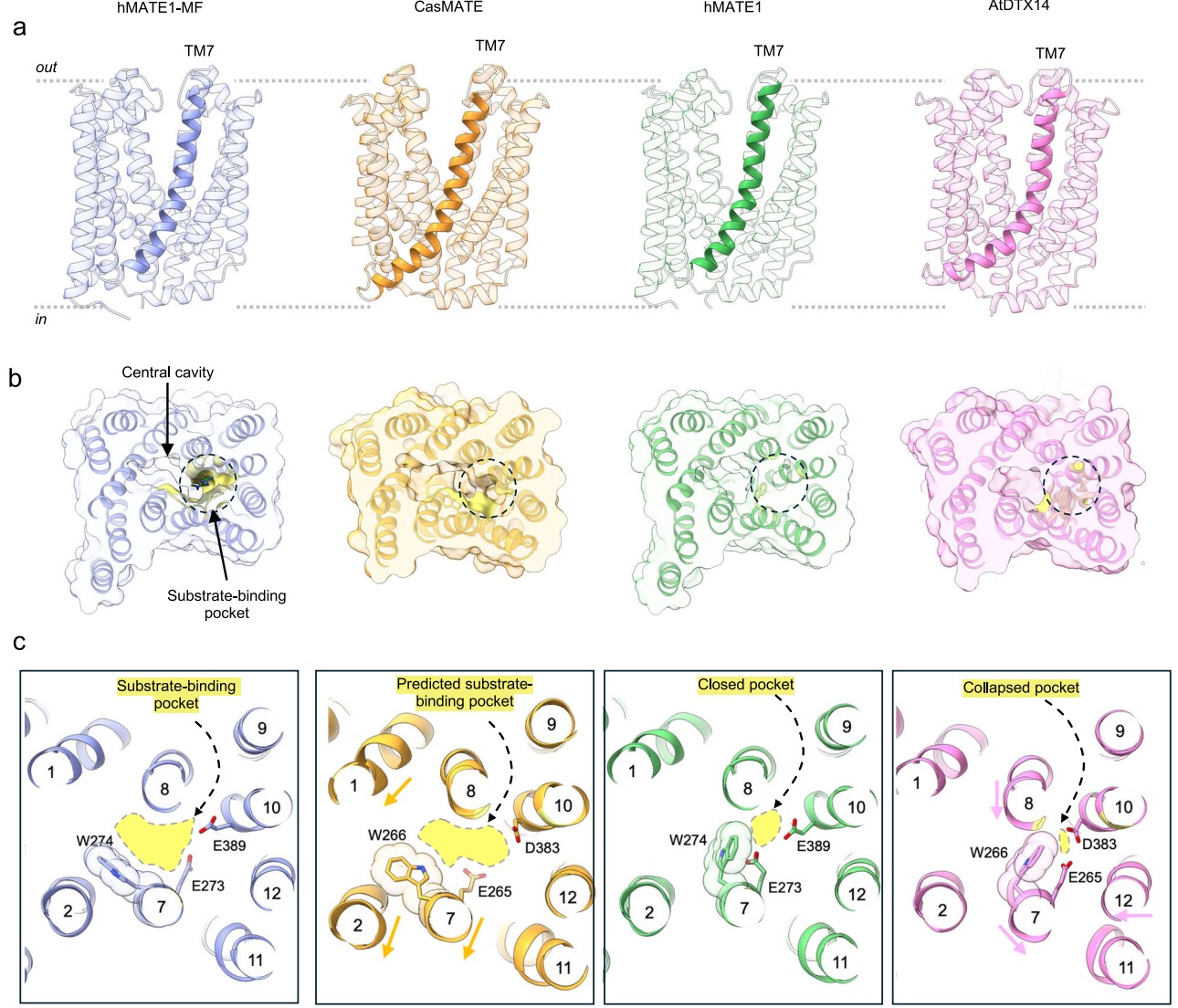

**Fig. 7 | Comparison of eukaryotic MATE structures. a** Ribbon representation of the structures of human MATE1 in apo and metformin-bound states (this study), plant MATEs CasMATE and AtDTX14 in apo and proposed H⁺-bound state respectively. **b** View from the extracellular side of the substrate binding pocket (colored in yellow). **c** A close-up view of the substrate binding pocket showing the opening and closing of the pocket caused by a conformational change of W274 in hMATE1, as well as collapsing of the pocket due to the movement of TM7 and W266 in plant MATEs. Conformational changes of the TM helices with respect to hMATE1-MF structure are shown as arrows.

Y277. The smaller size of metformin, and its high mobility in the pocket still allows MPP to be transported until metformin is present in large excess.

## Discussion

Through our cryo-EM structures and functional assays, we have successfully determined and validated the substrate binding pocket of hMATE1. The pocket is located in the C-lobe and is negatively charged. The conserved acidic residue pair E273 and E389 is critical for transport activity, as shown by our MPP uptake assays and consistent with previous studies on other hMATE1 substrates[22,30]. Their strong conservation among eukaryotic MATEs and their location in the binding pocket suggest proton binding at these residues, facilitating the proton-coupled antiport mechanism.

Despite previous functional studies pointing to a putative substrate binding pocket, structural validation has been lacking. Published eukaryotic structures include those of CasMATE and AtDTX14[21,22] which share ~32% amino acid sequence identity with hMATE1 (Fig. 7). No substrate binding was observed in these structures, but it was

predicted to be in the C-lobe, similar to our observation for hMATE1, and lined by the conserved residues W266 (W274 in hMATE1), E265 (E273 in hMATE1) and D383 (E389 in hMATE1) (Fig. 7c). Different crystallization conditions revealed a notable structural difference in TM7. In the CasMATE structure solved at pH 7.1, TM7 displays a 25° kink around residue C263, whereas in AtDTX14 at pH 5, the kink increases to 54° at the same position (Supplementary Fig. 13). This movement of TM7, accompanied by a conformational change of W266, collapses the predicted substrate binding pocket. As a result, the two glutamates are positioned within hydrogen bonding distance of each other, which suggests protonation of these residues. In the CasMATE structure, these two acidic residues are further apart at 6.5 Å and the tryptophan is oriented towards the central cavity. This creates a bigger pocket in the C-lobe, which allows for the binding of the substrate (Fig. 7c). The substrate-bound structure of hMATE1 closely resembles the conformation observed in the CasMATE structure. In this state, TM7 exhibits a 32° kink and the two conserved glutamates, E273 and E389, are positioned 4.6 Å apart (Supplementary Fig. 13). W274 is oriented towards the central cavity, allowing substrate

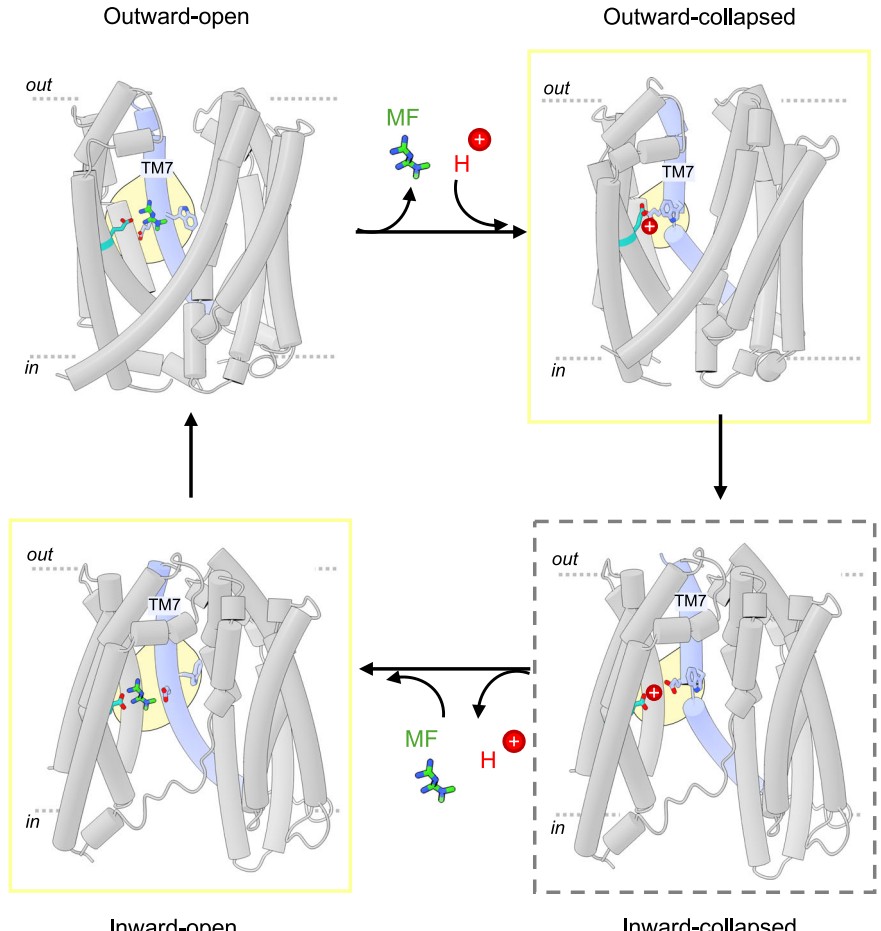

**Fig. 8 | Proposed transport mechanism of hMATE1 transporter.** Schematic diagrams illustrate the conformational changes of hMATE1 during the H⁺/substrate antiport. TM8 was hidden for clarity. Outward-open state corresponds to the hMATE1-MF structure from this study, with the substrate binding pocket colored in yellow. Outward-collapsed state was homology-modelled based on the AtDTX14 structure (PDB ID: 5y50). Inward-open state was homology-modelled based on the inward-facing PfMATE structure (PDB ID: 6fhz). Inward-collapsed is proposed based on the other models. Yellow box indicates homology-modelled structures. Dashed box indicates predicted.

binding. In contrast, in the apo structure of hMATE1, the W274 adopts a conformation that closes the pocket from the central cavity. Here, the pocket is closed rather than collapsed as it is not accompanied by a shift in TM7.

Similar conformations of CasMATE and hMATE1 structures are likely due to the same pH of the study (pH 7.4), at which both acidic residues are deprotonated and not interacting. While AtDTX14 probably represents an H⁺-bound state. The apo and substrate-bound structures of eukaryotic MATE transporters determined to date reveal an outward-open conformation. The only inward-facing conformation for the MATE family has been solved for archaeal PfMATE (PDB ID: 6fhz). Based on this structure, we have modelled an inward-facing hMATE1 using Swissmodeller.

Taken together, these structures allow us to propose a transport mechanism for hMATE1 (Fig. 8). In this proposed model, hMATE1 substrate-bound structures represent the outward-open conformation. A substrate (e.g. metformin) binds in the C-lobe cavity between TM7 and TM10, forming polar interactions with both glutamates (E273 and E389). A proton is likely to bind to E273 or E389, resulting in a conformational change of TM7 that collapses the C-lobe cavity and enables the formation of the hydrogen bond between the two glutamates. The movement of TM7 is likely to be accompanied by a conformational change of W274, which may help to shield the glutamate from the solvent-exposed central cavity and facilitate its protonation. The collapse of the C-lobe cavity is important to prevent rebinding of the substrate from the extracellular side. The rest of the helices rearrange to adopt an inward-facing conformation (Inward-collapsed, state not resolved). Tryptophan likely reorients towards the central cavity, leading to proton release from the glutamates. TM7 then undergoes a conformational change that pulls the two glutamates apart. This creates a bigger pocket to accommodate substrate binding from the intracellular side (Inward-open state), which is eventually released to the extracellular side.

## Methods

### Construct design and protein expression

Human MATE1 (Uniprot ID: Q96FL8) sequence was codon-optimized for expression in human cells (GeneArt, ThermoFisher) and purchased from GenScript. The synthetic gene was cloned into pcDNA5 vector containing 3C-cleavable N-terminal Flag-eYFP tag. pcDNA5 construct was used for tetracycline-inducible stable cell line generation (Flp-In™ T-REx™ 293 Cell Line, Thermo Fisher Scientific), following the manufacturer's guidelines. Cells were grown and maintained in Dulbecco's Modified Eagle Medium (DMEM, Gibco) supplemented with 10% fetal bovine serum (FBS, Thermo Fisher Scientific), 100 µg/mL streptomycin, 100 units/mL penicillin (Thermo Fisher Scientific) at 37 °C with 5% CO₂ under humidified conditions. Protein expression was induced with 1 µg/mL tetracycline (Sigma) at 37 °C. After 48 h of expression, cells were harvested and stored at −80 °C.

## hMATE1-mutant generation

Human MATE1 synthetic gene was cloned into a modified pUC57 vector containing 3C-cleavable N-terminal Flag-eYFP tag. This construct was used for transient transfection in HEK293-T cells. hMATE1 mutants were generated by site-directed mutagenesis PCR according to QuikChange protocol (Agilent). The primers used are listed in the Supplementary Data 1.

## Fluorescence microscopy

HEK293 cells stably expressing hMATE1 (24 h after induction) were gently resuspended in the fresh cell maintenance buffer and seeded on 6-well glass bottom plates (ibidi) at a density of $0.5 \times 10^5$ cells/well and cultured overnight. 48 h after induction, cells were incubated with uptake buffer (145 mM NaCl, 1 mM CaCl$_2$, 0.5 mM MgCl$_2$, 3 mM KCl, 5 mM D-glucose, and 10 mM Tris pH 8.2) containing 0.5 μM DAPI at 37 °C for 20 min. The fluorescent images of DAPI and YFP were directly visualized by using Nikon Eclipse Ti-E inverted microscope. Image analysis was conducted in Fiji ImageJ. DAPI uptake was quantified by performing a separate experiment, where the corresponding Fab was added at 9 μM to each well and incubated for 45 min. Then DAPI was added and incubated for 8 min. Reaction was stopped with addition of the cold uptake buffer and rinsed three times. Cells were lysed with 0.1% SDS and fluorescence intensity (360 nm excitation/460 nm emission) was measured (BioTek plate reader).

## Cell-based transport experiments

Stable cell line expressing hMATE1 was seeded onto a 24-well plate at a density of 100,000 cells/well previously coated with poly-D-lysine. Cells were grown for 24 h until they were ~80% confluent and then induced with 1 μg/ml tetracycline. For assays with mutants, HEK293-T cells were transiently transfected with 1 μg DNA of mutant plasmid and 2.5 μg of branched Polyethylenimine (Polysciences). As a control, HEK293-T cells transfected with empty vector (not coding for hMATE1) were included in the experiment.

Following 24 h of expression, media was removed and the cells were washed two times with 300 μl of pre-warmed Uptake Buffer (UB; 145 mM NaCl, 1 mM CaCl$_2$, 2 mM K$_2$HPO$_4$, 0.5 mM MgCl$_2$, 3 mM KCl, 5 mM D-glucose, and 25 mM HEPES pH 8.5). Uptake was started by the addition into each well of 250 μl of UB containing a total of 1 μM of non-radiolabeled and radio-labeled $^3$H-MPP mixture, at a molar ratio of 1:80 hot to cold. For the competition assays, non-radiolabeled cimetidine and metformin were added to each well in increasing concentrations. The uptake assay was stopped after 8 min by aspirating the UB and immediately washing the cells three times with 500 μl of ice-cold UB. Cells were solubilized by adding 250 μl lysis buffer (2% Triton X-100 and 1 M NaOH) and left to lyse for 1 h. Total protein concentration was measured in triplicates per each condition using the BCA assay according to the manufacturer's manual. The lysate was transferred to the scintillation tubes (4 mL Vial Ultima Gold; Perkin Elmer) and 2 ml of scintillation liquid was added. After 1 h incubation intracellular radioactivity was measured in the liquid scintillation counter.

The plasma membrane expression levels of each mutant protein were normalized to the expression level of the wild-type hMATE1 (Supplementary Fig. 12). For normalization, membrane protein fraction was extracted from 0.3 g of cells for each mutant using Plasma Membrane Protein Extraction kit (abcam®, ab65400). Briefly, cells were lysed with Homogenize Buffer and after low-speed centrifugation ($700 \times g$) and high-speed centrifugation ($10,000 \times g$) total cellular membrane protein was extracted. Plasma membrane protein was further isolated from the upper phase after mixing with Upper Phase solution and Lower Phase solution and centrifuging the mixture at $1000 \times g$. Same volume of each mutant plasma membrane extract was loaded on SDS-PAGE gel and hMATE1 specific band was detected by

Sapphire Imager (Azure Biosystems). Band intensity was quantified with ImageJ for each mutant.

## hMATE1 protein purification

Cell pellet was thawed and resuspended in a lysis buffer containing 40 mM HEPES pH 7.4, 150 mM NaCl, 10% (v/v) glycerol, 2 μg/mL DNaseI (Roche), 1 μg/mL soybean trypsin inhibitor (Sigma) and cOmplete protease inhibitor cocktail (Roche). It was lysed by 25 strokes in a dounce homogenizer. Cell lysate was solubilized with 1% DDM (n-dodecyl-β-d-maltopyranoside, Anatrace), 0.2% (w/v) CHS (cholesteryl hemisuccinate, Anatrace) for 90 min. The solubilized lysate was ultracentrifuged at $100,000 \times g$ in a Type 45-Ti rotor (Beckman) for 40 min. The supernatant was incubated with pre-equilibrated M2 Anti-FLAG Affinity gel (Millipore) for 2 h with gentle agitation. The resin was washed two times with ten column volumes (CV) of purification buffer containing 40 mM HEPES, pH 7.4, 150 mM NaCl, 5% glycerol, 0.02% DDM, and 0.004% CHS, followed by elution with two CV of wash buffer supplemented with 0.3 mg/ml Flag peptide and incubated for 3 h.

## Nanodisc reconstitution

Brain polar lipid extract (Avanti) was mixed with cholesterol (Avanti) in 4:1 ratio. The lipid mixture was solubilized with 1% DDM 0.2% CHS and 20 min sonication and added to purified hMATE1. After 5 min incubation at 4 °C purified membrane scaffold protein (MSP1D1) was added to the mixture and incubated for an additional 20 min at 4 °C. The reconstitution ratio for nanodisc of 1:5:85 (hMATE1:MSP1D1:lipids) was used. Bio-Beads were pre-washed with HBS (25 mM HEPES, pH 7.4, 150 mM NaCl) and added for 4 h incubation. To remove empty nanodiscs, hMATE1 reconstituted into nanodisc was incubated with 2 ml of NHS-activated Sepharose (cytiva) coupled to an anti-GFP nanobody for 2 h and washed three times with 10 CV of HBS. It was further eluted from the resin by overnight incubation with 3 CV of wash buffer containing 3C protease at a 1:50 wt/wt ratio. MATE1_Fab 3 or MATE1_Fab 6 was added in 1.2-fold molar excess to hMATE1 reconstituted into nanodisc and incubated for 30 min prior to size-exclusion chromatography step (TSK-gel G3000) in HBS buffer.

## Phage display selection

For phage display selection, hMATE1 was reconstituted into nanodiscs as described above using biotinylated MSP1D1. Before nanodisc assembly, MSP1D1 was chemically biotinylated using EZ-Link NHS-PEG4-Biotin (Thermo Scientific) according to reference[31]. Efficient biotinylation was confirmed using a pull-down assay with streptavidin(SA)-coated paramagnetic beads (Promega). Biopanning was performed using the Fab Library E[32] at 4 °C using published protocols[31]. The selection buffer was 25 mM HEPES pH 7.4, 150 mM NaCl supplemented with 1% (w/v) BSA. In the first round, 300 nM of target was immobilized on 250 μl SA magnetic beads. Then, 100 μl of a phage library E were added to the target bound to the SA beads and incubated for 30 min. The resuspended beads containing bound phages were washed extensively and then used to infect log phase *E. coli* XL1-Blue cells. Phages were amplified overnight in 2xYT media with 100 μg/ml ampicillin and $10^9$ p.f.u./ml of M13-KO7 helper phage. To increase the stringency of selection, five additional rounds of biopanning were performed with decreasing target concentrations (Round 2: 150 nM, Round 3: 75 nM, Round 4: 75 nM, Round 5: 25 nM, Round 6: 25 nM). For each round, the amplified phage pool from each preceding round was used as the input. For rounds 2–6, biopanning was performed semi-automatically using a Kingfisher magnetic beads handler (Thermo Fisher Scientific). To reduce the presence of non-specific binders, the phage pools for rounds 2–5 were precleared with 100 μl of streptavidin particles. For all rounds we used non-biotinylated empty nanodiscs without hMATE1 as soluble competitor in excess (3 uM in round 1 and 2 uM from rounds 2–6) to eliminate binders binding to the nanodiscs without the target. In rounds 2–6, bound phage particles were eluted

with 1% Fos-choline-12 prepared in selection buffer. This specific elution strategy enriches for binders targeting the membrane protein instead of enriching the phage pool with nanodisc binding clones.

## Single-point phage ELISA

The ELISA experiments were performed at 4 °C in 96-well plates coated with 50 μl of 2 μg/ml neutravidin in $Na_2CO_3$ buffer, pH 9.6 and subsequently blocked by 1.0% BSA in PBS. A single-point phage ELISA was used to rapidly screen the binding of the obtained Fab fragments displayed on phage. Colonies of *E.coli* XL1-Blue harboring phagemids from 5th and 6th rounds of selection were inoculated directly into 500 μl of 2xYT broth supplemented with 100 μg/ml ampicillin and M13-KO7 helper phage. The cultures were grown overnight at 37 °C in a 96-deep-well block plate. The ELISA buffer was identical to that used in selection. The experimental wells in the ELISA plates were incubated with 50 nM hMATE1 in biotinylated nanodiscs in ELISA buffer for 15 min. Wells containing biotinylated empty nanodiscs without hMATE1 were used to identify clones binding to empty nanodiscs. Overnight culture supernatants containing Fab phage were diluted 10-fold in ELISA buffer. The diluted phage supernatants were then transferred to ELISA plates that were pre-incubated with biotinylated target and washed with ELISA buffer. The ELISA plates were incubated with the phage for another 15 min and then washed with ELISA buffer. The washed ELISA plates were incubated with a 1:1 mixture of mouse anti-M13 monoclonal antibody (cat: 27-9420-01, GE, 1:5000 dilution in ELISA buffer) and peroxidase conjugated goat anti-mouse IgG (cat: 115-035-003, Jackson Immunoresearch, 1:5000 dilution in ELISA buffer) for 30 min. The plates were washed again, developed with TMB substrate and then quenched with 1.0 M HCl, and the absorbance at 450 nm was determined. The background binding of the phage was monitored by the absorbance from the control wells in which no target or empty nanodiscs were immobilized.

## Sequencing, cloning, expression and purification of Fab fragments

From phage ELISA, clones (selected based on a high ratio of ELISA signal of target binding to background) were sequenced at the DNA Sequencing Facility at the University of Chicago. Seven unique clones were obtained. These were sub-cloned in pRH2.2, an IPTG inducible vector for expression of Fabs in *E. coli*. *E. coli* C43 (Pro+) cells were transformed with sequence-verified clones of Fab fragments in pRH2.2[33]. Fab fragments were grown in TB autoinduction media with 100 μg/ml ampicillin overnight at 30 °C. Harvested cells were kept frozen at −80 °C until use. Frozen pellets were re-suspended in PBS supplemented with 1 mM PMSF. The suspension was lysed by ultra-sonication. The cell lysate was incubated at 65 °C for 30 min followed by centrifugation. The supernatant was filtered through 0.22 μm filter and loaded onto a HiTrap Protein L 5-mL column pre-equilibrated with lysis buffer (20 mM HEPES buffer, pH 7.5, 500 mM NaCl). The column was washed with 10 column volumes of lysis buffer followed by elution of Fab fragments with elution buffer (100 mM acetic acid). Fractions containing protein were directly loaded onto a Resource S 1-mL column pre-equilibrated with buffer A (50 mM sodium acetate, pH 5.0) followed by washing with 10 column volumes wash with buffer A. Fab fragments were eluted with a linear gradient 0–50% of buffer B (50 mM sodium acetate, pH 5.0, 2.0 M NaCl). Affinity and ion-exchange chromatography were performed using an automated program on ÄKTA explorer system. Purified Fabs were dialyzed overnight against 20 mM HEPES, pH 7.4, 150 mM NaCl. The quality of purified Fab fragments was analyzed by SDS–PAGE.

## Multipoint protein ELISA for $EC_{50}$ determination

Multipoint ELISA was performed at 4 °C to estimate the affinity of the Fabs to hMATE1 in nanodics. 50 nM of target immobilized on a neutravidin coated ELISA plate was incubated with 3-fold serial dilutions of the purified Fabs starting from 4 μM for 20 min. The plates were washed, and the bound target-Fab complexes were incubated with a secondary HRP-conjugated Pierce recombinant protein L (cat: 32420, Thermofisher, 1:5000 dilution in ELISA buffer) for 30 min. The plates were again washed, developed with TMB substrate and quenched with 1.0 M HCl, and absorbance ($A_{450}$) was determined. To determine the affinities, the data were fitted in a dose response sigmoidal function in GraphPad PRISM and $EC_{50}$ values were calculated.

## Cryo-EM sample preparation

Purified hMATE1 reconstituted into nanodisc and in complex with MATE1_Fab 3 or MATE1_Fab 6 and anti-Fab nanobody at a concentration of 0.6-1 mg/ml was added to different compounds at the final concentration of 1.5 mM MPP, 1.2 mM cimetidine and 10 mM metformin. A separate sample was prepared without the addition of any compounds and represents the apo state.

For cryo-EM grid freezing Vitrobot Mark IV (ThermoFisher Scientific) was used. A volume of 3.5 μL of the sample was applied to a glow-discharged Cu 300-mesh R1.2/1.3 grid (Quantifoil). The grid was blotted for 3 or 3.5 s under 100% humidity at 4 °C and then plunge-frozen into ethane-propane mixture.

## Cryo-EM data collection and processing

Data was collected on a Titan Krios 300 kV (Thermo Fisher Scientific) equipped with a Gatan K3 detector and GIF BioQuantum. Movies were acquired with EPU2 software (Thermo Fisher Scientific) at a nominal magnification of 130,000×, in super-resolution counting mode and in CDS mode. The movies were binned twice in EPU2 and the final pixel size was 0.65 Å/pix. The defocus ranged from −0.6 to −2.2 μm, with a total dose of 42.5 e−/Å² spread over 50 frames with a total exposure time of 2.5 s.

Collected movies were processed with CryoSPARC (v4). The processing details for each dataset are presented in the Supplementary Figs. 2–6. General pipeline for each dataset was the following: Firstly, the movies were motion-corrected using Patch Motion Correction and CTF was estimated using Patch CTF estimation. Micrographs with an overall resolution better than 5 Å were selected for further processing. Particles were picked using blob picker, extracted and binned 4× and then 2D classified for at least three times. Best 2D classes, representing different particle views, were used for creating four ab-initio models, which were used further for heterogeneous refinement. Best 3D volume was selected and was subjected to non-uniform refinement. The particles were then re-extracted (unbinned to 0.65 Å/pix) and used for another round of non-uniform refinement. Afterwards, particles were subjected to local refinement using the mask that excludes nanodisc signal and the constant part of the Fab. From there on, the volume was used for template picking. The new particles were 2D classified and used for heterogeneous refinement. Best class was selected and used for non-uniform refinement, local refinement, global and local CTF refinement. For hMATE1-metformin dataset template picking was skipped, as the map reached high resolution at earlier steps. Final resolutions of the 3D maps were hMATE1-Fab6-apo (2.95 Å), hMATE1-Fab3-apo (3.7 Å), hMATE1-MF (2.3 Å), hMATE1-MPP (3.15 Å), hMATE1-CMT (3.3 Å).

## Model building and refinement

The final refined EM maps were used for model building in Coot[34]. Alphafold[35,36] structure of hMATE1 (Uniprot ID Q96FL8) was used for model building of apo hMATE1. The model was fitted into EM map, manually adjusted and refined in PHENIX[37]. For drug-bound states, apo model was manually adjusted and refined in PHENIX and the corresponding ligand was docked. Cimetidine, MPP and metformin geometry restrictions were generated from their corresponding SMILE codes using eLBOW[38].

## Molecular dynamics simulations

The MATE1 structure was embedded in a $100 \times 100$ Å membrane bilayer composed of 1-palmitoyl-2-oleoyl-sn-glycero-3-phosphocholine (POPC) lipids (Supplementary Table 2). The system was solvated with a 25 Å water layer and neutralized with $Cl^-$ ions. The final unit cell dimensions were 100 Å $\times$ 100 Å $\times$ 100 Å. System assembly was performed using the Leap module from AmberTools (version 23.6). The ff19SB force field[39] was used for the protein and lipid21[40] parameters were applied to model the lipid bilayer. The TIP3P[41] water model was used for solvation.

The parameters for Metformin were obtained from a previous study[42]. MPP and Cimetidine were parametrized with Antechamber[43] and Parmchk2 (AMBER23 suite), using GAFF2 for atom typing and AM1-BCC for partial charge assignment. Five simulation systems were constructed: (i) a system containing a single metformin molecule bound to MATE1, (ii) a system with two metformin molecules, (iii) a system with a single MPP molecule; (iv & v) two systems, representing one of the possible binding poses of cimetidine.

Simulations were performed using OpenMM (version 8.2.0)[44], following the minimization and equilibration protocols provided by CHARMM-GUI[45]. Periodic electrostatic interactions were calculated using the Particle Mesh Ewald (PME) method. Force switching was employed for van der Waals interactions, with a switching distance set at 9 Å. Langevin dynamics with a friction coefficient of $1 \, ps^{-1}$ were used to maintain a constant temperature of 303.15 K. Constant pressure of 1 atm was maintained with Monte Carlo barostat[46]. Hydrogen bond constraints were used to achieve a 2 fs time step. Each system was simulated in four independent replicas, with trajectories extending up to 600 ns.

Trajectory analysis was carried out using VMD (version 1.9.4a57)[47] and in-house scripts written in tcl and python. To evaluate ligand mobility within the binding pocket, the root mean square deviation (RMSD) of each ligand was computed relative to its experimental configuration across replicas. Additionally, distances from the center of the carboxylic group of residue E389 to a selected atom for each ligand were measured. For metformin, the terminal guanidine nitrogen atom (N5) was chosen, while for MPP, the nitrogen atom was selected. For cimetidine, two atoms were considered to evaluate different functional groups: N6 in the imidazole ring and N17 in the cyanoguanidine moiety.

## Figure preparation

Figures were prepared using ChimeraX v.1.7[48] and GraphPad Prism v.10[49]. Chemical structures were prepared using ChemDraw (Revvity Signals Software, Inc). Membrane elements used in Fig. 1 were adapted from Servier Medical Art (https://smart.servier.com/), licensed under CC BY 3.0 (https://creativecommons.org/licenses/by/3.0/). Protein structure illustration in Fig. 1 was generated from our own cryo-EM data using the pdb2vector tool (https://bioicons.com/pdb2vector/).

## Reporting summary

Further information on research design is available in the Nature Portfolio Reporting Summary linked to this article.

## Data availability

The atomic coordinates have been deposited in the Protein Data Bank (PDB) under accession numbers 9R1G (hMATE1-apo), 9R1F (hMATE1-MF), 9R1O (hMATE1-CMT), 9R1E (hMATE1-MPP). The three-dimensional cryo-EM density post-processed, masked maps have been deposited in the Electron Microscopy Data Bank (EMDB) under accession number EMD-53508 (hMATE1-apo), EMD-53507 (hMATE1-MF), EMD-53489 (hMATE1-CMT) and EMD-53506 (hMATE1-MPP).

Data related to molecular dynamics simulations are available at Zenodo with the identifier 15836460. Source data are provided with this paper.

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

## Acknowledgements

We thank Nevena Srejic (ETH Zurich) for technical support in fluorescent images acquisition. We thank the Scientific Center for Optical and Electron Microscopy (ScopeM) at ETH Zürich for technical support. This research was supported by SNSF NCCR TransCure (grant 51NF40-185544) and SNSF grant 214834 to K.P.L., SNSF grant PCEFP3_194606 to T.L. and National Institutes of Health grant GM117372 to A.A.K.

## Author contributions

K.P.L. and K.R. conceived the study. K.R. performed cloning, protein expression, purification, functional assays, sample preparation for Fab selection and for cryo-EM studies. J.K. performed cryo-EM data collection. K.R. performed the cryo-EM analysis and model building. G.P. and T.L. performed and analyzed the molecular dynamics simulations. S.M., L.R. and J.H. performed Fab selection, ELISA screening, Fab expression and purification. A.A.K. supervised synthetic antibody generation. K.R. and K.P.L wrote the manuscript with input from all authors.

## Competing interests

The authors declare no competing interests.
