## [Transparent Peer Review file · Nature Communications]

Structural basis of drug recognition by human MATE1 transporter

Corresponding Author: Dr Kaspar Locher

Version 0:

Reviewer comments:

Reviewer #1

(Remarks to the Author)

The articles describe the cryo-EM structure of human multidrug and toxin extrusion protein (hMATE) and its functional assay, with results that are scientifically intriguing. These findings enhance the field's understanding of hMATE; however, uncertainties persist in the modeling, which require further validation.

Major issues :

1. In Figure 3a and page 5, the four structures require further validation. In the apo structure, an endogenous density corresponding to unknown ligands is expected to exist in the substrate-binding site. Additionally, the primary conformation of residue TRP274 should align with that observed in the other three structures. The density assigned to W274-1 is highly likely to originate from the unknown endogenous substrate density, which also overlaps with the densities of MF and CMT. As a result, the orientation of MF and CMT is insufficiently clear to confidently support their modeled placement. The densities do not align well with the chemical structures of MF and CMT, necessitating caution when interpreting their placement and orientation.
2. Since Fb stabilizes the conformation of hMATE, it may induce a state with even weaker or no binding of MF and CMT. This should also be considered by the authors.
3. Even when excluding density considerations, the assignment of CMT in Figure 4 does not align with that of MF. The guanidine group is in an opposite position.
4. The π - π interaction between MPP and TRP274 is ignored by the author.
5. The MD simulations for MF and CMT are may also unreliable due to the modelling problem.

Minor points

1. Line 101 : Is there any reference for the MATE fold? What is the difference between the MATE fold and the MFS fold?
2. Line 117 : Concentrations approximately 10-fold above the reported Km of these drugs are far for enough in cryo-EM.
3. In Figures S2–S6, "9'496 movies" should be corrected to "9,496 movies," and other spelling issues should also be addressed.

Reviewer #2

(Remarks to the Author)

This is quite a nice paper IMO - MATE1 is relevant, there hasn't been a structure, now we have (a cryo-EM) one, which is obviously of relevance.

The work contains an apo as well as complexed structures and mutation work, as well as MD simulations of bound ligands to investigate binding poses. The latter is to a good extent mostly a hypothesis, but that is intrinsic to the nature of the work. I cannot judge the experimental methods in all details (we did more computational work on MATE1 before), but from my side I cannot see anything that stands out as problematic.

The authors could include a discussion of previous MATE1 homology models and proposed binding modes as well, and compare previous models to the experimental work - on the other hand this is not really necessary (and might be more due to my own interest), so I certainly do not insist on that and the focus on this structure makes the work more succinct.

The paper is quite well-written by the way - to-the-point, avoiding the waffle of many manuscripts (likely written with AI help) I get to review these days, the authors are hence thanked for their writing style as well.

Good work from what I can see and worth publishing.

Reviewer #3

(Remarks to the Author)

This manuscript by Romane et al reports cryo-EM structural characterization of the drug recognition mechanism by human MATE1, which belongs to the multidrug and toxin extrusion transporter family and plays an essential role in mediating the final excretion step of drugs and endogenous metabolites into urine and bile. Due to the low molecular weight of the MATE1 protein and its lack of soluble domains, structural determination of such protein posed significant challenges. In this manuscript, the authors use Fab to facilitate structure determination and successfully resolved four distinct MATE1 structures: the apo state and three ligand-bound states with metformin, cimetidine and MPP. These structures appear to be of high quality, and provide a clear model for drug recognition. The associated MD simulations and functional assays validate the structural model. This is an important study regarding the understanding of structural basis of hMATE1 substrate specificity and the drug transport mechanism. I only have some minor suggestions, mostly directed at making the presentation clearer for readers:

1. Line 7, "MPP". Since this drug is first mentioned in line 7 in this manuscript, its full name should be used here, not in line 63.
2. Line 87, the "CDR-H3". Some readers may not be familiar with the term "CDR." Please define it at the first occurrence in the manuscript.
3. The authors identified an additional density in the hMATE1 structure, which they assigned to a cholesterol molecule. We recommend that the authors discuss the potential functional roles of cholesterol in this protein.
4. Line 386, "Hepes". The authors used "Hepes" here but "HEPES" elsewhere in the manuscript. Please standardize the term to ensure consistency.
5. In Fig. 4, the image for panel (d) is positioned too far from its label, which may cause confusion.
6. In the legend of Fig.6a, the "S.E." should be "S.E.M."
7. In Figure 8, the authors should clearly label the meanings of the yellow solid-line box and black dashed-line boxes, ideally indicating their definitions in the figure legends.
8. In Supplementary Fig.2-6, the authors should add a scale bar in the panel (b) and add an extra panel to show the particle orientation distributions of the final refinement structure.
9. In Supplementary Fig.2-6, Supplementary Table 1 and the main text, the authors used apostrophes as thousand separators (e.g., 1'000'000). While this format is common in countries like Germany or Switzerland, it does not align with international standards for English scientific writing. Please revise all instances to use commas as thousand separators (e.g., 1,000,000).
10. In Supplementary Fig. 8, the YFP fluorescence is displayed as a grayscale image, whereas Fig. 1c shows the typical pseudo-colored representation of YFP. Could the authors clarify why grayscale was chosen for this figure? If intentional, we recommend adding a brief note in the figure legend to explain this stylistic choice (e.g., emphasizing the membrane expression level among different hMATE1 mutants).

Version 1:

Reviewer comments:

Reviewer #1

(Remarks to the Author)

The authors have addressed most of my concerns, with only two minor points remaining:

For the cryo-EM density, the contour level should be labeled in the figures, including Figure 3. In Figure 6C, the uptake assay results for Y306F appear questionable. With only four data points and one outlier, the reported IC₅₀ value of 0.7 μM may not be reliable (in contrast, the wild-type IC₅₀ is 2.7 μM).

We thank the reviewers for their insightful comments. In the following, reviewer comments are in black, while our responses / actions are colored red for clarity.

Reviewer #1 (Remarks to the Author)

The articles describe the cryo-EM structure of human multidrug and toxin extrusion protein (hMATE) and its functional assay, with results that are scientifically intriguing. These findings enhance the field's understanding of hMATE; however, uncertainties persist in the modeling, which require further validation.

Major issues :

1. In Figure 3a and page 5, the four structures require further validation. In the apo structure, an endogenous density corresponding to unknown ligands is expected to exist in the substrate-binding site. Additionally, the primary conformation of residue TRP274 should align with that observed in the other three structures. The density assigned to W274-1 is highly likely to originate from the unknown endogenous substrate density, which also overlaps with the densities of MF and CMT. As a result, the orientation of MF and CMT is insufficiently clear to confidently support their modeled placement. The densities do not align well with the chemical structures of MF and CMT, necessitating caution when interpreting their placement and orientation.

We fully agree with the reviewer that caution has to be taken for cryo-EM densities in MDR transporters. We believe we have exercised great caution when interpreting cryo-EM maps, and we agree that adding more views of the binding pocket will be helpful in demonstrating the quality of the EM density maps. We have modified Figure 3 and have added the another Supplementary Figure (Supplementary Fig. 7), which is shown here:

This supplementary figure shows that the EM density for metformin and cimetidine is very strong. It also clearly shows that the densities of metformin and cimetidine differ in shape and size from the density observed in the apo state.

We acknowledge the reviewer's point that several multidrug transporters have been shown to present density of endogenous compound in the apo state, as in the case of OCT1 (Zeng YC *et al*, Nat Comm, 2023). However, in our case the presence of extra density is accompanied by mobility of W274 in the apo state (alternates between W274-1 and W274-2), which becomes ordered in the substrate-bound state (W274-2). This suggests that W274 is stabilized through interactions with bound ligands, while in the absence of ligand it remains flexible.

We believe that our MD simulations and mutational analysis strongly support our ligand placement and offer the best possible validation. To avoid redundancy, we expand on this point in our response to Comment 5.

2. Since Fab stabilizes the conformation of hMATE, it may induce a state with even weaker or no binding of MF and CMT. This should also be considered by the authors.

We agree that the Fab binding has the potential to influence substrate interaction, an observation that is in line with many published transporter structure studies. The combination of structural data of non-inhibitory (MATE1_Fab 3) and inhibitory Fab (MATE1_Fab 6) strongly support that both Fabs recognize essentially the same outward-facing conformation of hMATE1 (Figure 2). The difference between the two Fabs, as shown by functional assays, is that MATE1_Fab 3 permits substrate transport, whereas MATE1_Fab 6 inhibits it. We interpret this such that Fab6 traps the outward-open conformation (preventing transport) but does not interfere with substrate binding. This is supported by our cryo-EM data of three ligand-bound structures. We have clarified this point in the revised manuscript to better reflect this interpretation. Lines 101-105:

"This confirmed that MATE1_Fab 6 binding does not distort the protein structure but instead locks a native conformational state, supporting the physiological relevance of the Fab-bound structures. Since both Fabs trap the same conformation, we infer that both Fabs are compatible with substrate binding, while MATE1_Fab 6 blocks the substrate release."

3. Even when excluding density considerations, the assignment of CMT in Figure 4 does not align with that of MF. The guanidine group is in an opposite position.

The reviewer is correct that cimetidine and metformin do not bind the same way. This is to be expected and in line with findings made with other multidrug transporters. The evidence from our MD simulations suggest that our cimetidine placement is correct because unlike the alternative pose (where cimetidine is found to diffuse away from the binding pocket quickly, see Movie 3), the cimetidine molecule built in our structure remained stable. To clarify the findings, we have added a panel to Supplementary Fig. 9 to show the alternative pose of cimetidine.

4. The π - π interaction between MPP and TRP274 is ignored by the author.
We thank reviewer for this point, we now mention the π - π interaction between MPP and Trp274 in the text and have indicated it in the Figure 4.

5. The MD simulations for MF and CMT are may also unreliable due to the modelling problem.

As indicated above, we don't think there is a "modelling problem." Instead, we believe that MD simulations provide valuable insights into the dynamic behavior and stability of the ligands within the binding pocket. We considered multiple ligand placements for both metformin and cimetidine to evaluate various orientations / poses. We have added panels to Supplementary Fig. 9 to illustrate these alternative poses.

For cimetidine, we initiated simulations from two distinct starting models with different orientations. Of these, only one orientation remained stably bound throughout the simulation, while the other rapidly dissociated (Movie 3), strongly suggesting that the stable orientation is the most plausible binding mode.

For metformin, we examined six initial binding poses: one derived from manual fitting and refinement and five from docking with Swissdock. All of the simulations (in four replicates) resulted in metformin sampling the same conformational space within the binding pocket. Notably, metformin shows more pronounced dynamics and motion within the binding pocket and could even reorient during the simulation. Therefore, the binding pose we propose is not absolute, but a probable one based on the putative interactions, that were validated by the mutational analysis. All of this is indicated in the text. It aligns well with findings made and published with other multidrug transporters, where substrates can adopt multiple poses. This is a key feature of such transporters.

Minor points

1. Line 101 : Is there any reference for the MATE fold? What is the difference between the MATE fold and the MFS fold?

We thank the reviewer for this suggestion. In the revised manuscript, we have now clarified the structural distinction between the MATE and MFS folds and included appropriate references (lines 53-57)

2. Line 117 : Concentrations approximately 10-fold above the reported K_m of these drugs are far for enough in cryo-EM.

We selected ligand concentrations approximately 10-fold above the reported K_m values as a starting point, balancing the need for sufficient occupancy with the requirements imposed on maximum concentration before EM grid quality was affected. At higher concentrations of compounds such as cimetidine, which is dissolved in DMSO, EM grid quality could be impaired. As we observed clear ligand density at these concentrations in our cryo-EM maps, we proceeded with this range for structure determination.

3. In Figures S2–S6, "9'496 movies" should be corrected to "9,496 movies," and other spelling issues should also be addressed.

We have corrected the number format in the main text, Supplementary Figures and Table.

Reviewer #2 (Remarks to the Author):

This is quite a nice paper IMO - MATE1 is relevant, there hasn't been a structure, now we have (a cryo-EM) one, which is obviously of relevance.

The work contains an apo as well as complexed structures and mutation work, as well as MD simulations of bound ligands to investigate binding poses. The latter is to a good extent mostly a hypothesis, but that is intrinsic to the nature of the work. I cannot judge the experimental methods in all details (we did more computational work on MATE1 before), but from my side I cannot see anything that stands out as problematic.

The authors could include a discussion of previous MATE1 homology models and proposed binding modes as well, and compare previous models to the experimental work - on the other hand this is not really necessary (and might be more due to my own interest), so I certainly do

not insist on that and the focus on _this_ structure makes the work more succinct.

The paper is quite well-written by the way - to-the-point, avoiding the waffle of many manuscripts (likely written with AI help) I get to review these days, the authors are hence thanked for their writing style as well.

Good work from what I can see and worth publishing.

We thank the reviewer for their positive assessment of our work and for their suggestions. We have now introduced some previous findings from homology models about the proposed binding poses in the lines 64-67.

Reviewer #3 (Remarks to the Author)

This manuscript by Romane et al reports cryo-EM structural characterization of the drug recognition mechanism by human MATE1, which belongs to the multidrug and toxin extrusion transporter family and plays an essential role in mediating the final excretion step of drugs and endogenous metabolites into urine and bile. Due to the low molecular weight of the MATE1 protein and its lack of soluble domains, structural determination of such protein posed significant challenges. In this manuscript, the authors use Fab to facilitate structure determination and successfully resolved four distinct MATE1 structures: the apo state and three ligand-bound states with metformin, cimetidine and MPP. These structures appear to be of high quality, and provide a clear model for drug recognition. The associated MD simulations and functional assays validate the structural model. This is an important study regarding the understanding of structural basis of hMATE1 substrate specificity and the drug transport mechanism. I only have some minor suggestions, mostly directed at making the presentation clearer for readers:

1. Line 7, "MPP". Since this drug is first mentioned in line 7 in this manuscript, its full name should be used here, not in line 63.

We agree and have revised the text to introduce the full name of 1-methyl-4-phenylpyridinium (MPP) at its first mention in line 7.

2. Line 87, the "CDR-H3". Some readers may not be familiar with the term "CDR." Please define it at the first occurrence in the manuscript.

We thank the reviewer for this suggestion. We have now defined CDR in lines 94-95 of the revised manuscript as follows: *"For MATE1_Fab 6 most of the interactions are mediated by the longest loop, CDR-H3 (complementarity-determining region 3 of the heavy chain), which interacts with both the N- and C-lobe, while CDR-H1 and CDR-H2 further stabilize the complex (Fig. 2c)."*

3. The authors identified an additional density in the hMATE1 structure, which they assigned to a cholesterol molecule. We recommend that the authors discuss the potential functional roles of cholesterol in this protein.

We thank the reviewer for this suggestion. The cholesterol density observed in our hMATE1 structures likely originates from the nanodisc reconstitution process, which included addition of brain polar lipid extract supplemented with cholesterol. To our knowledge, the functional role of cholesterol in hMATE1 has not been previously investigated. We believe that additional studies are necessary that would include hMATE1 reconstitution into defined lipid systems like

proteoliposomes to explore the functional contribution of cholesterol. We have clarified this further in text, lines 121-122:

“To our knowledge, the role of cholesterol in MATE1 has not been previously investigated, and any potential functional relevance remains to be determined.”

4. Line 386, “Hepes”. The authors used "Hepes" here but "HEPES" elsewhere in the manuscript. Please standardize the term to ensure consistency.

Done.

5. In Fig. 4, the image for panel (d) is positioned too far from its label, which may cause confusion.

We have adjusted the Fig. 4 to ensure that the image for panel (d) is properly aligned with its label.

6. In the legend of Fig.6a, the “S.E.” should be “S.E.M.”.

Done.

7. In Figure 8, the authors should clearly label the meanings of the yellow solid-line box and black dashed-line boxes, ideally indicating their definitions in the figure legends.

We have updated Figure 8 to include labels for the yellow solid-line box and black dashed-line boxes.

8. In Supplementary Fig.2-6, the authors should add a scale bar in the panel (b) and add an extra panel to show the particle orientation distributions of the final refinement structure.

We have updated Supplementary Fig. 2–6 to include scale bars in panel (b). Additionally, we have added a new panel (f) to each figure, showing the particle orientation distribution from the final refinement.

9. In Supplementary Fig.2-6, Supplementary Table 1 and the main text, the authors used apostrophes as thousand separators (e.g., 1'000'000). While this format is common in countries like Germany or Switzerland, it does not align with international standards for English scientific writing. Please revise all instances to use commas as thousand separators (e.g., 1,000,000).

We have corrected the number format in the main text, Supplementary Figures and Table.

10. In Supplementary Fig. 8, the YFP fluorescence is displayed as a grayscale image, whereas Fig. 1c shows the typical pseudo-colored representation of YFP. Could the authors clarify why grayscale was chosen for this figure? If intentional, we recommend adding a brief note in the figure legend to explain this stylistic choice (e.g., emphasizing the membrane expression level among different hMATE1 mutants).

The YFP fluorescence in Supplementary Fig. 8 is shown in grayscale to allow for clearer visual comparison of membrane expression levels across different hMATE1 mutants. We have added a brief explanation in the figure legend for clarification: “eYFP signal shown as grayscale to emphasize the membrane expression level among different hMATE1 mutants.”

REVIEWERS' COMMENTS

Reviewer #1 (Remarks to the Author):

The authors have addressed most of my concerns, with only two minor points remaining:

1) For the cryo-EM density, the contour level should be labeled in the figures, including Figure 3.

We have not included explicit contour levels in the figures because these values are arbitrary for cryo-EM visualization and depend on map normalization, mask softness, and other parameters. To ensure clarity, we instead follow the standard practice in the field, where ligand density is shown together with surrounding aminoacid residues. This minimizes the risk of overinterpretation.

We note that recent (2025) cryo-EM studies published in *Nature Communications* presenting cryo-EM densities of bound ligands do not specify contour levels in their figure legends either, demonstrating that our approach is the current best practice:

<https://www.nature.com/articles/s41467-025-61226-x>

<https://www.nature.com/articles/s41467-025-60480-3>

<https://www.nature.com/articles/s41467-025-57956-7>

In Figure 6C, the uptake assay results for Y306F appear questionable. With only four data points and one outlier, the reported IC_{50} value of 0.7 μM may not be reliable (in contrast, the wild-type IC_{50} is 2.7 μM).

To quantify reliability from the Y306F dataset, we report here the IC_{50} with a 95% confidence interval derived from a profile-likelihood fit in Prism. The fitted IC_{50} is 0.7 μM , with a 95% CI of 0.41-1.46 μM . For wild-type, the fitted IC_{50} is 2.7 μM (95% CI: 1.62–4.62 μM). The confidence intervals for Y306F and wt do not overlap, supporting the conclusion that Y306 presents a left shift of the IC_{50} relative to wild-type. For clarity we provide below the Y306F inhibition curve, with dashed lines indicating the confidence interval. Removal of the outlier would lead to IC_{50} of 1 μM , which would not alter the conclusion of the left shift of the IC_{50} relative to wild-type.